# Nonregular Physical Activity and Handgrip Strength as Indicators of Fatigue and Psychological Distress in Cancer Survivors

**DOI:** 10.3390/curroncol32050289

**Published:** 2025-05-21

**Authors:** Ilaria Pepe, Alessandro Petrelli, Francesco Fischetti, Carla Minoia, Stefania Morsanuto, Livica Talaba, Stefania Cataldi, Gianpiero Greco

**Affiliations:** 1Department of Translational Biomedicine and Neuroscience (DiBraiN), University of Bari ”Aldo Moro”, 70124 Bari, Italy; ilaria.pepe@uniba.it (I.P.); alessandro.petrelli@uniba.it (A.P.); gianpiero.greco@uniba.it (G.G.); 2Department of Neurosciences, Biomedicine and Movement Sciences, University of Verona, 37129 Verona, Italy; 3Hematology Unit, IRCCS Istituto Tumori “Giovanni Paolo II”, 70124 Bari, Italy; c.minoia@oncologico.bari.it; 4Department of Education and Sport Sciences, Pegaso Telematic University, 80143 Naples, Italy; stefania.morsanuto@unipegaso.it (S.M.); stefania.cataldi@unipegaso.it (S.C.); 5Department of Surgical Pathology, University of Pisa, 56126 Pisa, Italy; livica.talaba@unipi.it

**Keywords:** physical inactivity, muscle function, anxiety–depressive symptoms, cancer-related fatigue, cancer survivorship, mental and physical well-being

## Abstract

Background: Cancer survivors who do not engage in regular physical activity often experience persistent psychological distress and fatigue, which can significantly impact their quality of life. While handgrip strength (HGS) is recognized as an indicator of overall health and physical resilience, the combined role of HGS and physical inactivity in predicting psychological distress and fatigue in this population remains unclear. This study aimed to examine the relationships between self-reported physical inactivity, HGS, and psychological distress, specifically depressive symptoms, anxiety, and cancer-related fatigue (CRF), in physically inactive cancer survivors. Methods: This cross-sectional study included 42 physically inactive cancer survivors (mean age = 63.2 years, SD = 8.96) recruited from the Cancer Institute (IRCCS) in Bari, Italy. Physical inactivity was quantified based on self-reported weekly physical activity minutes, with all participants engaging in less than 150 min per week. The participants underwent HGS assessment and completed validated psychological measures, including the Beck Depression Inventory (BDI), the State-Trait Anxiety Inventory (STAI-Y1 and STAI-Y2), and the Fatigue Severity Scale (FSS). Results: Bivariate correlations were examined via Spearman’s rank correlation coefficients, and multiple linear regression analyses were performed to identify independent predictors of psychological distress and fatigue, adjusting for covariates such as age, sex, cancer type, and time since treatment completion. Both lower HGS and greater physical inactivity were significantly correlated with greater depressive symptoms (HGS: ρ = −0.524, *p* < 0.001; physical inactivity: ρ = −0.662, *p* < 0.001), greater fatigue severity (HGS: ρ = −0.599, *p* < 0.001; physical inactivity: ρ = −0.662, *p* < 0.001), and increased trait anxiety (HGS: ρ = −0.532, *p* < 0.001; physical inactivity: ρ = −0.701, *p* < 0.001). No significant associations were found between physical inactivity or HGS and state anxiety (*p* > 0.05). Multiple regression analyses confirmed that both HGS and physical inactivity independently predicted depressive symptoms (HGS: β = −0.435, *p* = 0.009; physical inactivity: β = −0.518, *p* = 0.002), trait anxiety (HGS: β = −0.313, *p* = 0.038; physical inactivity: β = −0.549, *p* < 0.001), and fatigue (HGS: β = −0.324, *p* = 0.033; physical inactivity: β = −0.565, *p* < 0.001), even after adjusting for covariates. Low physical activity and reduced muscle strength independently predict psychological distress and fatigue in cancer survivors. Conclusions: These findings highlight the potential exacerbating role of physical inactivity in both physical and psychological vulnerability, underscoring the need for interventions promoting regular exercise. Integrating strength assessments and structured physical activity programs may be key strategies in survivorship care to improve mental well-being and overall quality of life.

## 1. Introduction

Cancer is a major public health problem worldwide. Although advances in treatments have led to a reduction in mortality, projections indicate that by 2030, 22.1 million people will be living with a history of cancer [1]. In addition to a rise in new diagnoses as a population ages and grows, this significant increase also indicates longer survival as a result of advancements in early detection and treatment [2]. Chemotherapy, radiation, and/or biological therapy are commonly administered to cancer patients multiple times, with active surveillance in addition to therapy. These treatments can induce chronic pain/negative impacts in cancer survivors, reducing physical activity and leading to physical deconditioning, which ultimately compromises recovery and quality of life [3].

Guidelines from the American College of Sports Medicine and the American Cancer Society recommend that cancer survivors engage in at least 150 min of moderate-intensity or 75 min of vigorous-intensity aerobic activity per week or an equivalent combination of moderate to vigorous physical activity (MVPA) weekly [4,5]. However, many survivors fail to meet these guidelines, and physical inactivity is prevalent; self-reported physical inactivity levels (failing to achieve 150 min of moderate-intensity physical activity or 75 min of vigorous-intensity physical activity per week) range from 29% of the population in high-income countries to 15% in low-income countries [6].

Physical inactivity and sedentariness are highly prevalent and are associated with muscle atrophy, reduced aerobic capacity, and decreased strength [7]. A lack of regular exercise not only worsens physical deconditioning but also exacerbates psychological symptoms. In fact, according to recent studies, almost 50% of cancer survivors still feel exhausted [8,9], whereas 30% to 40% suffer from mental health conditions, including depression, anxiety, and various adjustment or psychosomatic disorders [10,11], even after completing treatment. However, physical activity is a safe, feasible, and relatively inexpensive nonpharmacological alternative for the management of psychological symptoms in cancer survivors [12]. Increasing physical activity levels may reduce symptoms and thus improve quality of life during survivorship [13].

Several risk factors for fatigue, depression, and anxiety have been identified in the literature in breast cancer survivors following treatment [14,15,16,17]. These symptoms may be influenced by demographic factors [18], treatment modalities [19], time since treatment completion, pain, sleep disturbances, menopausal symptoms [20], psycho-emotional factors, physical activity levels [21], and physical functional impairments, including poor handgrip strength (HGS) [22].

As a direct indicator of muscle function, HGS is a significant predictor of muscle mass and total physical strength, reflecting multiple aspects of physical function [23]. Reduced HGS has been linked to lower levels of physical activity, as inactivity contributes to progressive muscle weakening, which in turn exacerbates functional decline [24].

A lower HGS is associated with a greater incidence of anxiety–depressive symptoms [25,26] and a lower quality of life [22,27], indicating that muscle weakness could be a direct result of the combined effects of the psychological state and cancer-related fatigue (CRF) on physical function [28]. Assessing whether HGS is associated with a combined symptomatic profile rather than individual symptoms is particularly relevant in cancer survivors, where fatigue, depression, and anxiety frequently coexist and contribute to a self-perpetuating cycle of physical inactivity, muscle deterioration, and worsening psychological well-being [28,29,30].

Evidence suggests that reduced muscle strength is associated with both physical and psychological vulnerability, reinforcing the idea that deconditioning is a key factor in the persistence of symptoms [31,32]. If HGS and self-reported physical inactivity serve as reliable indicators of overall symptom burden, they could provide an accessible, noninvasive screening approach for identifying high-risk individuals and guiding targeted interventions, including structured rehabilitation and psycho-oncological support [33].

Critically, while previous studies have examined HGS in general cancer populations, few have specifically focused on physically inactive survivors, a group particularly prone to muscle deconditioning and psychological distress. Given that inadequate physical activity may further exacerbate both physical and psychological vulnerability, it is essential to evaluate the impact of both muscle strength and inactivity on mental health outcomes in this population. Therefore, the present study aims to (i) examine the associations among physical inactivity (defined as engaging in < 150 min of self-reported physical activity per week), HGS, and the overall symptom profile in cancer survivors who do not meet American College of Sports Medicine (ACSM) physical activity recommendations. Specifically, we investigate their relationships with depressive symptoms, anxiety, and cancer-related fatigue (ii) to determine whether lower HGS and increasingly fewer minutes of self-reported physical activity are independently associated with greater psychological distress and fatigue severity, reinforcing the hypothesis that both reduced muscle strength and inadequate physical activity contribute to heightened psychological symptom burden. On the basis of prior evidence, we hypothesize that, among physically inactive cancer survivors, (i) lower HGS is significantly associated with higher levels of depression, increased fatigue, and heightened anxiety; (ii) fewer self-reported minutes of weekly physical activity are significantly associated with greater psychological distress and fatigue severity; and (iii) both HGS and self-reported physical activity independently predict psychological symptom burden, suggesting that reduced muscle strength and insufficient physical activity contribute to mental health deterioration in cancer survivors.

## 2. Materials and Methods

### 2.1. Sample and Study Design

This cross-sectional observational and analytical study was conducted at the Cancer Institute (IRCCS) Giovanni Paolo II Cancer Institute in Bari, Italy, a specialized center for cancer treatment and rehabilitation. The study included 42 cancer survivors who did not practice regular physical activity, with a mean age of 63.2 years (SD = 8.96). Recruitment and data collection were carried out in January 2025 during routine oncological follow-up visits and rehabilitation appointments. Eligible participants who met the inclusion criteria were invited to participate in a single study session, during which they completed validated psychological assessments and underwent HGS measurement. Ethical approval was obtained from the Institutional Ethics Committee of Bari (protocol code 0324886|30/12/24), and all participants provided written informed consent prior to study enrollment. The reporting of this study follows the Strengthening the Reporting of Observational Studies in Epidemiology (STROBE) statement (Appendix A).

### 2.2. Participants and Sampling

Cancer survivors were recruited via a consecutive sampling approach, whereby all eligible patients attending the Bari Cancer Institute during the recruitment period were systematically screened for inclusion. This method facilitated the selection of a representative sample of cancer survivors attending routine follow-up visits, thereby minimizing selection bias. The eligibility criteria required participants to have completed primary cancer treatment at least six months prior to enrollment, ensuring that acute treatment effects did not confound the study outcomes. Patients who self-reported nonadherence to physical activity guidelines, defined as engaging in less than 150 min of moderate-intensity aerobic activity, 75 min of vigorous-intensity aerobic activity per week, or an equivalent combination of MVPA, were included in the study. Additional inclusion criteria included age ≥ 18 years and the ability to independently complete the study procedures. The exclusion criteria included neurological or musculoskeletal disorders affecting HGS, severe cognitive impairment, ongoing participation in a structured physical rehabilitation program, or the use of medications with significant neuromuscular effects that could interfere with HGS assessment. To determine the appropriate sample size, a priori power analysis was performed via G*Power 3.1 [34]. On the basis of a multiple linear regression model, assuming a medium effect size (f^2^ = 0.25), an alpha level of 0.05, and a statistical power (1-β) of 0.80, the minimum required sample size was 38 participants. To account for potential missing data, 42 cancer survivors were ultimately enrolled in the study.

### 2.3. Measurement

During the clinical interview, each patient was informed about the study’s aims and methods and provided written consent to participate. After socioanamnestic and clinical data, including self-reported physical activity expressed in weekly minutes, a key study variable alongside HGS, were collected, participants underwent a comprehensive psychological and motor assessment. Standardized measures were used to evaluate depressive symptoms, anxiety, and fatigue, whereas HGS testing was performed to assess muscle strength. Both self-reported physical activity and HGS were analyzed as continuous variables to investigate their independent and combined associations with psychological distress and fatigue severity.

#### 2.3.1. Muscle Strength

The handgrip strength test is a reliable and practical method for evaluating the maximum voluntary strength of both the extrinsic and intrinsic muscles of the hand. In this study, grip strength was measured via a mechanical Smedley hand dynamometer (GIMA, Milan, Italy). The participants were seated in a straight-backed chair with their feet flat on the floor. Their shoulders were kept close to the body (0° flexion), their elbows were bent at a 90° angle, and their forearms were in a neutral position. They were instructed to grip the dynamometer as firmly as possible and maintain the squeeze for 5 s. The test was performed three times for each hand, with a 30-s rest between each attempt. A 60-s break was given before switching to the other hand. At the end of the test, the average of the three measurements for each hand (dominant and nondominant), recorded in kilograms (kg), was used for further analysis.

#### 2.3.2. Psychological and Fatigue Assessment

Depressive symptoms were assessed via the Beck Depression Inventory (BDI), a 21-item self-report questionnaire designed to measure the severity of depressive symptoms experienced over the past week [35,36]. Each item represents a specific symptom of depression (e.g., sadness, pessimism, feelings of failure, dissatisfaction, guilt, self-criticism, irritability, sleep disturbances, fatigue, and loss of appetite) and is rated on a four-point scale (0–3), with higher scores reflecting greater symptom severity. The total BDI score ranges from 0 to 63, with established cut-off values for classifying depression severity as minimal (0–9), mild (10–16), moderate (17–29), or severe (30–63) [35].

Anxiety levels were assessed via the State-Trait Anxiety Inventory (STAI-Y), a 40-item self-report questionnaire that distinguishes between state anxiety (STAI-Y1)—temporary, situation-dependent anxiety—and trait anxiety (STAI-Y2)—a more stable, enduring tendency toward anxious responses [37]. Each subscale consists of 20 items rated on a four-point Likert scale (1 = not at all to 4 = very much), with total scores ranging from 20 to 80, where higher scores indicate greater anxiety severity. The STAI-Y1 assesses transient emotional states related to a specific moment (e.g., feeling tense, worried, or nervous in response to a particular situation), whereas the STAI-Y2 evaluates a more generalized anxiety disposition, reflecting a long-standing proneness to experiencing anxiety across different contexts.

Fatigue severity was assessed via the Fatigue Severity Scale (FSS), a 9-item questionnaire designed to evaluate the impact of fatigue on daily functioning [38]. The participants rated each item on a 7-point Likert scale (1 = strongly disagree to 7 = strongly agree), with higher scores indicating a greater burden of fatigue. The total FSS score is obtained by summing all nine items (range 9–63), with a total score ≥ 36 indicating clinically significant fatigue. The FSS is particularly relevant in cancer research, as it assesses both the physical and cognitive consequences of fatigue, making it a useful tool for evaluating cancer-related fatigue (CRF), one of the most common and debilitating long-term effects in cancer survivors.

### 2.4. Additional Variables and Potential Confounders

To control for potential confounding factors, demographic and clinical variables, including age (continuous variable, years), sex (male/female), marital status, tumor type, and time since treatment completion (continuous variable, months), were recorded. Since age, sex, cancer type, and time since treatment have been associated with differences in depression, anxiety, and fatigue levels among cancer survivors, these variables were included in the analysis to adjust for possible effects on psychological well-being and fatigue. The cancer type variable was dichotomized as “breast cancer” vs. “other cancer types” because of the high prevalence of breast cancer survivors in the sample. Similarly, marital status was classified as “married” or “unmarried” to ensure adequate group size and allow their inclusion in regression models by dummy variables.

### 2.5. Statistical Analysis

All the statistical analyses were conducted via the IBM SPSS Statistics 26.0 program (IBM, Chicago, IL, USA), with significance set at *p* < 0.05 (two-tailed tests). Before performing the primary analyses, assumption checks were conducted to ensure the validity of the statistical models. Shapiro–Wilk tests and Q–Q plots were used to assess the normality of residuals, whereas scatterplots and Levene’s test were applied to check for linearity and homoscedasticity. Multicollinearity diagnostics were performed using variance inflation factor (VIF) thresholds of <10 to ensure that there was no collinearity among the predictors.

Before addressing the primary research questions, preliminary analyses were conducted to assess whether psychological distress (depressive symptoms and anxiety) and fatigue differed across key covariates, including age, sex, marital status, cancer type, and time since treatment completion. Given the nonnormal distribution of the psychological outcome variables, Mann–Whitney U tests were performed to compare differences in BDI, STAI-Y1, STAI-Y2, and FSS scores between categorical variables (i.e., sex, marital status, and cancer type), whereas Spearman’s rank correlation coefficients (ρ) were used to examine associations between psychological outcomes and continuous variables (i.e., age and time since treatment completion).

To evaluate the associations between self-reported minutes of physical activity per week, HGS, psychological distress (depressive symptoms and anxiety), and fatigue, Spearman’s rank correlation coefficients (ρ) were computed, given the nonparametric nature of the data. This analysis provided an initial assessment of the relationships between physical inactivity, HGS, and the symptomatologic profile. To examine the predictive role of physical inactivity and HGS in psychological symptoms and fatigue, separate multiple linear regression models were constructed with the BDI, STAI-Y1, STAI-Y2, and FSS scores as dependent variables. Physical inactivity and HGS were entered as the primary predictors, whereas age, sex, marital status, cancer type, and time since treatment completion were included as covariates. A backwards stepwise method was applied, allowing nonsignificant covariates to be iteratively removed from the models to identify the most parsimonious predictors of psychological distress and fatigue. The final models reported standardized beta coefficients (β) and adjusted R² values, which were used to assess the proportion of variance explained by the predictor variables. Collinearity diagnostics (tolerance and variance inflation factor (VIF)) were performed to ensure model stability. Additionally, residual plots were examined to confirm the appropriateness of the model assumptions.

## 3. Results

### 3.1. Characteristics of the Participants

The sociodemographic and clinical characteristics of the participants are summarized in Table 1. A total of 42 individuals (9 males and 33 females), all of whom completed the psychological assessment and upper limb muscle strength evaluation, were included. The participants had a mean age of 63.2 ± 8.96 years (mean ± SD). Most of the participants were married (88.1%). Breast cancer was the most prevalent diagnosis, accounting for 64.3% of the cases, whereas other cancer types represented 35.7% of the sample. The average time since treatment completion was 17.80 ± 9.65 months. The mean scores for the psychological and physical variables assessed in the study are also reported in Table 1. The participants had an average FSS score of 36.19 ± 12.9, indicating moderate to high fatigue levels. Anxiety scores differed between the state and trait components, with a mean STAI-Y1 score of 35.10 ± 6.3 and a mean STAI-Y2 score of 50.71 ± 11.2, suggesting greater levels of trait anxiety than state anxiety. The mean BDI score was 22.57 ± 10.1, corresponding to an overall moderate level of depressive symptoms. The mean HGS was 22.07 kg ± 7.3. Additionally, the mean self-reported physical activity was 57.86 ± 32.33 min per week, confirming that all participants were classified as physically inactive according to established guidelines.

No significant differences were observed in fatigue levels, depressive symptoms, or anxiety scores between participants based on marital status (pFSS = 0.792, pSTAI-Y1 = 0.236, pSTAI-Y2 = 0.732, and pBDI = 0.880, respectively), cancer type (pFSS = 0.968, pSTAI-Y1 = 0.810, pSTAI-Y2 = 0.875, and pBDI = 0.545, respectively), or sex (pFSS = 0.380, pSTAI-Y1 = 0.566, pSTAI-Y2 = 0.506, and pBDI = 0.364, respectively). Similarly, no significant associations were observed between age and psychological or fatigued outcomes or between age and time since completion of treatment (all *p* > 0.05) (see Table 2). 

### 3.2. Associations of Self-Reported Weekly Physical Activity Minutes and HGS with Psychological and Fatigue Symptomatology

Significant negative correlations were detected between self-reported weekly physical activity minutes and depressive symptoms (ρ = −0.662, *p* < 0.001), and trait anxiety (ρ = −0.701, *p* < 0.001), and fatigue (ρ = −0.662, *p* < 0.001), whereas no significant relationships were detected between self-reported weekly physical activity minutes and state anxiety (ρ = −0.132, *p* = 0.406) (Table 3). Additionally, self-reported weekly physical activity minutes was significantly associated with lower HGS (ρ = 0.428, *p* = 0.005).

Similarly, lower HGS was correlated with greater depressive symptoms (ρ = −0.524, *p* < 0.001) and fatigue (ρ = −0.599, *p* < 0.001), as well as higher levels of trait anxiety (ρ = −0.532, *p* < 0.001). No significant correlation was found between HGS and state anxiety (ρ = −0.158, *p* = 0.317) (Table 3).

Spearman’s rho coefficients indicate moderate to strong associations among physical inactivity, diminished muscle strength, and psychological distress, with the most pronounced correlations observed for trait anxiety, depressive symptoms, and fatigue.

Multiple linear regression analyses were performed to further examine the predictive role of both physical inactivity and HGS on psychological distress and fatigue (Table 4). In these models, depressive symptoms, trait anxiety, state anxiety, and fatigue were entered as dependent variables, whereas self-reported weekly physical activity minutes and HGS were included as primary predictors, with age, sex, marital status, cancer type, and time since treatment completion included as covariates.

The adjusted R² values indicated that the models explained approximately 40–50% of the variance in depressive symptoms, trait anxiety, and fatigue, highlighting the substantial contribution of physical inactivity and muscle weakness to these outcomes.

Both self-reported weekly physical activity minutes and HGS were significant predictors of cancer-related fatigue (physical inactivity: β = −0.565, *p* < 0.001; HGS: β = −0.324, *p* = 0.033), trait anxiety (physical inactivity: β = −0.549, *p* < 0.001; HGS: β = −0.313, *p* = 0.038), and depressive symptoms (physical inactivity: β = −0.518, *p* = 0.002; HGS: β = −0.435, *p* = 0.009), indicating that both lower minutes of physical activity and reduced muscle strength contribute to greater psychological distress and fatigue. State anxiety was not significantly predicted by any of the explanatory variables (all *p* > 0.05). On the basis of tolerance and the VIF, the model had no multicollinearity problems (see Table 4).

## 4. Discussion

### 4.1. Key Results and Interpretation

This study investigated the relationships among self-reported weekly physical activity minutes, HGS, psychological symptoms, and fatigue in cancer survivors not engaging in regular physical activity as defined by the ACSM criteria. The results indicated that physical inactivity was significantly associated with increased levels of depressive symptoms, greater fatigue, and greater trait anxiety. Additionally, lower HGS was associated with the same psychological outcomes: greater depressive symptoms, increased fatigue, and heightened trait anxiety. However, no significant relationship was observed between self-reported weekly physical activity minutes or HGS and state anxiety.

These results support our hypothesis that, in physically inactive cancer survivors, both physical deconditioning (as indicated by reduced HGS) and inactivity itself contribute significantly to a greater burden of psychological distress and fatigue.

Symptoms of depression, anxiety, fear of recurrence, and fatigue are widespread among cancer survivors, who are physically inactive, often persist well after completion of treatment [39,40,41] and have a significant effect on quality of life [42,43,44]. Studies have consistently shown that these symptoms often coexist, contributing to the emotional distress and functional impairment of survivors [45,46,47].

Depressive symptoms and fatigue represent two of the most debilitating long-term consequences of cancer survivorship [48], as found in our study. Research indicates that depression affects up to 30–40% of survivors, with symptoms ranging from low mood and anhedonia to more severe impairments in daily functioning [49,50]. Similarly, cancer-related fatigue (CRF) is reported by more than half of survivors and often persists for years after treatment [51,52].

Anxiety, particularly trait anxiety, also plays a key role in the psychological distress experienced by cancer survivors [53]. High trait anxiety is associated with persistent worry, heightened stress responses, and difficulty returning to precancer levels of emotional well-being [54,55].

Survivors experiencing persistent fatigue, depressive symptoms, or heightened anxiety may be less likely to engage in physical activity [30,56], leading to progressive deconditioning and muscle weakness [57,58].

Many studies have highlighted the association between reduced muscle strength and psychological distress in cancer survivors, particularly in relation to depression [25,26] and quality of life [59,60]. However, to our knowledge, our study is novel in that it specifically examines the relationship between physical inactivity, HGS, and the broader symptom profile in a group that does not engage in regular physical activity, highlighting the unique interaction between inactivity, muscle deconditioning, and psychological distress.

Physical activity often decreases during survival, and the number of survivors meeting physical activity guidelines remains low [61], as found in our study. A substantial number of randomized controlled studies, reviews, and meta-analyses have revealed that physical inactivity is associated with fatigue, psychological distress, and worse overall quality of life in cancer survivors [30,40,62].

Physical inactivity has been associated with reduced levels of brain-derived neurotrophic factor (BDNF), a key regulator of neuroplasticity and mood stabilization [63]. Moreover, prolonged inactivity may contribute to dysregulation of the hypothalamic–pituitary–adrenal (HPA) axis, heightening stress responses and exacerbating mood disturbances [64]. Chronic inflammation, often observed in cancer survivors due to the lingering effects of treatment, may further amplify these associations, as proinflammatory cytokines have been implicated in both fatigue and depressive symptoms [65].

Psychosocial factors further contribute to this relationship. Physically inactive cancer survivors may experience greater social isolation [66], reduced engagement in enjoyable or meaningful activities [67], and a diminished sense of control over their health [68], which are all factors linked to increased depressive symptoms and lower quality of life. Additionally, inactivity is often associated with poor sleep quality [69], which may exacerbate both fatigue and mood disturbances. From a psychological perspective, physical inactivity can negatively impact self-efficacy and perceived control [70]. Cancer survivors who engage in regular physical activity often report feeling more empowered in managing their health [68], whereas those who remain inactive may struggle with feelings of helplessness and lack motivation, further perpetuating emotional distress [67].

A better understanding of the biological mechanisms underlying the associations of physical activity and sedentary behavior with fatigue can provide more information about the potential causal pathways of these associations [71].

Reduced physical activity leads to muscle atrophy, decreased bone density, and metabolic dysregulation [57,58,72], all of which contribute to fatigue and diminished physical functioning. Additionally, inactivity is linked to unfavorable changes in body composition, including increased body mass index (BMI) [73], which further exacerbates fatigue and decreases motivation for physical engagement [74].

CRF, whether stemming from treatment side effects, anemia, or underlying health conditions, can lead to further reductions in physical activity [75], initiating a self-perpetuating cycle of deconditioning and worsening symptoms [75,76]. Over time, this cycle reinforces both physiological and psychological consequences, making it increasingly difficult for individuals to break free from sedentary behaviors [77].

HGS, as an indicator of upper limb muscle function, has emerged as a potential marker of both physical resilience and psychological well-being in oncology patients practicing physical activity [22,27,60,73]. Our findings align with this evidence, demonstrating that lower HGS is significantly associated with greater fatigue, depressive symptoms, and trait anxiety in cancer survivors not engaging in regular physical activity. The observed relationships between lower HGS and increased depression and fatigue suggest that functional impairments may contribute to worsening psychological well-being, further exacerbating the burden of survivorship. This association may be partly attributed to systemic inflammation and neuroendocrine dysregulation, mechanisms known to contribute to both muscle deterioration and mood disorders [78,79,80]. Furthermore, dysregulation of the hypothalamic–pituitary–adrenal (HPA) axis, which leads to altered cortisol secretion, has been related to both mood disorders and reductions in muscular function [81,82,83,84]. Additionally, the strong association between low HGS and fatigue severity is consistent with previous findings suggesting that reduced muscle function reflects underlying mitochondrial dysfunction and energy metabolism impairments [85]. CRF is a multifactorial condition influenced by inflammatory cytokines, hormonal imbalances, and muscle deconditioning [86,87], all of which are interrelated with reduced HGS [88,89]. Further knowledge about the specific forms of psychological distress experienced by cancer survivors can be gained from the association between HGS and trait anxiety but not state anxiety. Trait anxiety reflects a stable predisposition toward anxiety, which has been linked to chronic stress, autonomic dysregulation, and reduced physiological resilience [90]. This aligns with our findings that individuals with lower muscle strength, potentially reflecting greater physical vulnerability, exhibit higher levels of trait anxiety. In contrast, state anxiety is more situational and context dependent, which may explain the absence of significant associations with both HGS and physical inactivity in this study. Unlike trait anxiety, which reflects a more stable predisposition to anxiety-related experiences, state anxiety fluctuates in response to immediate stressors, such as medical appointments, recent test results, or psychosocial challenges [91,92].

Given its transient nature, state anxiety may be less influenced by long-term physiological factors such as muscle strength or habitual physical inactivity. Moreover, while chronic inactivity has been linked to psychological distress, its impact may be more pronounced on persistent mood disturbances, such as depression or fatigue, rather than on anxiety states that are inherently dynamic and shaped by acute stressors. This distinction suggests that the absence of associations in our findings may reflect the differing temporal characteristics of state anxiety compared with more stable indicators of psychological and physical health [93].

The relationships among physical inactivity, HGS, depression, anxiety, and fatigue are likely mediated by a dysfunctional biopsychosocial cycle that perpetuates physical and psychological decline. Cancer survivors who experience high levels of anxiety and depression may have a reduced tendency to engage in physical activity, as in our sample [30,94,95], resulting in progressive deconditioning and loss of muscle function [96,97,98], which is reflected in decreased hand strength. The decline in muscle strength and overall physical capacity may, in turn, exacerbate psychological symptoms [25,26], reinforcing a vicious cycle of inactivity, decreased strength and worsening mental health. Increased depressive and anxiety symptoms lead to reduced motivation and energy to engage in physical activity [99,100]. Reduced minutes of physical activity lead to muscle disuse and decreased HGS, resulting in reduced physical resilience [72]; decreased muscle strength and overall functional capacity contribute to increased fatigue, decreased self-efficacy and worsened mental health, further reinforcing physical activity avoidance [22,101].

Given these findings, physical activity and HGS emerge as important parameters to assess in physically inactive cancer survivors, not only as markers of physical function and psychological well-being but also as modifiable factors that can be improved through targeted interventions. Unlike other clinical and psychological risk factors that are not modifiable, physical activity and HGS can be improved through the introduction of and participation in structured programs, particularly those incorporating resistance training and aerobic exercise [102,103].

Although our study focused on self-reported exercise practices, it is well established that knowledge and attitudes towards physical activity critically mediate behavioral engagement. Accordingly, structured programs should not only measure exercise practices but also assess patients’ exercise knowledge and attitudes, as these tools can identify educational needs and motivational barriers in cancer survivors [104]. Incorporating structured education and motivational support alongside practice-based interventions may therefore enhance adherence and long-term exercise adoption.

There is robust evidence demonstrating that physical activity interventions in cancer survivors lead to improvements in fatigue [105,106,107], depressive symptoms [21,108], and overall quality of life [109,110]. Resistance training, in particular, has been shown to increase muscle strength and endurance [111,112,113], which in turn may contribute to reductions in cancer-related fatigue and psychological distress [33,114,115]. Given that fatigue, depression, and anxiety are interconnected with physical function, improving muscle strength through exercise could provide a dual beneficial effect, both by enhancing HGS and breaking the dysfunctional cycle of inactivity and psychological burden. 

### 4.2. Strengths and Limitations

This study contributes to the literature by strengthening the evidence supporting the relationship between physical inactivity, muscle strength, and psychological well-being in cancer survivors not engaging in regular physical activity. Our results highlight that physical activity and HGS are simple, noninvasive, cost-effective and clinically relevant methodologies for identifying individuals at increased risk for psychological distress and fatigue. Unlike many clinical and psychological risk factors that cannot be modified, muscle strength and physical activity practices are modifiable through targeted interventions, highlighting their importance in cancer survivorship. Among all the covariates included in the analysis, physical inactivity and HGS were the only variables significantly associated with depressive symptoms, trait anxiety, and fatigue, emphasizing their distinct roles as determinants of physical and psychological health in this population. These findings suggest that their incorporation into the routine care of survivors could offer valuable clinical insights, allowing for the early identification of individuals at risk of persistent psychological and physical impairment.

Despite these strengths, several limitations must be recognized. The cross-sectional design precludes causal inferences, making it unclear whether physical inactivity and muscle weakness contribute to psychological distress or whether psychological distress leads to reduced physical function and activity. Longitudinal studies are needed to establish the directionality of these associations and to determine whether changes in muscle strength and activity practices over time correspond to changes in depressive symptoms, anxiety, and fatigue. In addition, no standardized questionnaires were used, but only self-reported values of leisure-time physical activity were collected, excluding those who followed the ACSM guidelines.

Although adjustments were made for key covariates such as age, sex, cancer type, and time since treatment, other unmeasured factors, including nutritional status and medication use, may have influenced the observed associations. Furthermore, the use of consecutive samples from a single institution and the small sample size may limit the generalizability of these results. Larger, multicenter studies with more diverse cohorts of cancer survivors are needed to confirm these results and strengthen their external validity.

## 5. Conclusions

This study reinforces the role of low physical activity and HGS as significant correlates of depressive symptoms, fatigue, and trait anxiety in cancer survivors, emphasizing that these associations were observed in a physically inactive population.

The lack of regular physical activity among participants suggests that reduced muscle strength may not only serve as a marker of psychological distress but also reflect the broader consequences of a sedentary lifestyle, which is known to negatively impact both musculoskeletal function and mental well-being. Despite some limitations, the associations observed demonstrate the detrimental effects of inactivity on physical and mental health.

Given that physical activity and muscle strength are modifiable factors, future research should explore whether targeted resistance training interventions can attenuate psychological distress and improve general well-being in physically inactive cancer survivors. These findings support the integration of routine muscle strength assessments into survivor care and increased leisure-time physical activity practices, not only as screening tools to identify those at increased risk for psychological distress but also as a modifiable goal for rehabilitation programs. Encouraging structured physical activity, particularly strength training, may be a key strategy to improve both physical resilience and mental health outcomes in this population.

## Figures and Tables

**Table 1 curroncol-32-00289-t001:** Demographic and clinical characteristics of cancer survivors not engaging in regular physical activity (n = 42).

Variables	Statistics Total Sample
Age (Mean, SD)	63.2 ± 8.96
Sex (n, %) Male Female	9 (21.4) 33 (78.6)
Marital status (n, %) Married Unmarried	37 (88.1) 5 (11.9)
Cancer type Breast cancer Other cancer types	27 (64.3) 15 (35.7)
Time since treatment (Mean, SD)	17.80 ± 9.65
FSS (Mean, SD)	36.19 ± 12.90
STAI-Y1 (Mean, SD)	35.10 ± 6.31
STAI-Y2(Mean, SD)	50.71 ± 11.20
BDI (Mean, SD)	22.57 ± 10.10
Self-reported weekly physical activity minutes (Mean, SD)	57.86 ± 32.33
HGS (Mean, SD)	22.07 ± 7.35

**Abbreviations:** FSS, Fatigue-Severity Scale; STAI-Y1, State-Trait Anxiety Inventory; STAI-Y2, State-Trait Anxiety Inventory; BDI, Beck Depression Inventory; HGS, Handgrip Strength Test; SD, standard deviation

**Table 2 curroncol-32-00289-t002:** Analysis of outcomes in relation to covariates of interest.

Variables	FSS	*p* Value	STAI-Y1	*p* Value	STAI-Y2	*p* Value	BDI	*p* Value
Age ^1^	−0.077	0.629	−0.076	0.634	−0.153	0.333	0.047	0.766
Sex ^2^								
Male	18.22	0.380	19.33	0.566	19.06	0.506	18.11	0.364
Female	22.39		22.09		22.17		22.42	
Marital status ^2^								
Married	21.31	0.792	20.66	0.236	21.72	0.732	21.61	0.880
Unmarried	22.90		27.70		19.90		20.70	
Cancer type ^2^								
Breast cancer	21.56	0.968	21.83	0.810	21.72	0.875	20.65	0.545
Other	21.40		20.90		21.10		23.03	
Time since treatment ^1^	−0.299	0.054	−0.126	0.427	-0.121	0.444	−0.102	0.522

**Notes:** ^1^ Spearman’s correlation coefficient; ^2^ means of ranks based on the Mann–Whitney test. **Abbreviations:** FSS, Fatigue-Severity Scale; STAI-Y1, State-Trait Anxiety Inventory; STAI-Y2, State-Trait Anxiety Inventory; BDI, Beck Depression Inventory.

**Table 3 curroncol-32-00289-t003:** Relationships between self-reported weekly physical activity minutes, HGS, anxious-depressive symptoms, and FSS.

	FSS	STAI-Y1	STAI-Y2	BDI	HGS
HGS					
Spearman correlation	−0.599 **	−0.158	−0.532 **	−0.524 **	1
Sig. (2-tailed)	0.000	0.317	0.000	0.000	-
N	42	42	42	42	42
Self-reported weekly physical activity minutes					
Spearman correlation	−0.662 **	-0.132	−0.701 **	−0.662 **	0.428 **
Sig. (2-tailed)	0.000	0.406	0.000	0.000	0.005
N	42	42	42	42	42

**Abbreviations:** FSS, Fatigue-Severity Scale; STAI-Y1, State-Trait Anxiety Inventory; STAI-Y2, State-Trait Anxiety Inventory; BDI, Beck Depression Inventory; HGS, Handgrip Strength Test; Sig. (2-tailed), significance (2-tailed); **Notes:** ** *p* < 0.01.

**Table 4 curroncol-32-00289-t004:** Multiple linear regressions adjusted for confounding factors with self-reported weekly physical activity minutes and HGS as the dependent variables and psychological distress as the predictor.

	FSS			BDI			STAI-Y1			STAI-Y2				
	*β*	*SE*	*p*	*β*	*SE*	*p*	*β*	*SE*	*p*	*β*	*SE*	*p*	*Tolerance*	*VIF*
Age	0.029	0.167	0.805	−0.136	0.14	0.279	−0.093	0.132	0.624	−0.158	0.144	0.179	0.893	1.12
Sex	−0.058	6.114	0.77	−0.199	5.12	0.351	−0.101	3.781	0.688	−0.110	5.279	0.579	0.31	3.23
Marital status	0.193	5.626	0.108	−0.13	3.874	0.308	−0.319	−0.058	0.751	0.172	3.987	0.151	0.868	1.152
Type of cancer	0.028	4.447	0.868	−0.270	3.547	0.403	−0.023	2.535	0.905	−0.190	3.832	0.261	0.429	2.33
Time since treatment	−0.055	0.174	0.676	0.151	3.724	0.297	−0.026	−0.005	0.98	−0.006	0.15	0.962	0.704	1.421
HGS	−0.324	0.339	0.033	−0.435	0.295	0.009	−0.898	−0.22	0.376	−0.313	0.22	0.038	0.56	1.786
Self-reported weekly physical activity minutes	−0.565	0.056	*p* < 0.001	−0.518	0.047	0.002	−0.278	0.04	0.189	−0.549	0.049	*p* < 0.001	0.602	1.66
	R^2^ = 0.597 Adjusted R^2^ = 0.515			R^2^ = 0.534 Adjusted R^2^ = 0.438			R^2^ = 0.120 Adjusted R^2^ = −0.061			R^2^ = 0.59 Adjusted R^2^ = 0.5158				

**Abbreviations:** FSS, Fatigue-Severity Scale; STAI-Y1, State-Trait Anxiety Inventory; STAI-Y2, State-Trait Anxiety Inventory; BDI, Beck Depression Inventory; HGS, Handgrip Strength Test; β, standardized coefficient; SE, standard error; *p*, *p* value; VIF variance inflation factor; R^2^, R-squared; Adjusted R^2^, Adjusted R-squared.

## Data Availability

The data presented in this study are available upon request from the first author. The data are not publicly available due to privacy concerns.

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
