# Peer review of "Nonregular Physical Activity and Handgrip Strength as Indicators of Fatigue and Psychological Distress in Cancer Survivors"

_curroncol, 2025, doi:10.3390/curroncol32050289_

Round 1
Reviewer 1 Report
Comments and Suggestions for Authors
We commend the authors for addressing an important and vulnerable population—cancer survivors experiencing physical inactivity. This manuscript explores the relationships between physical activity, muscle strength, psychological well-being, and fatigue, providing valuable insights into the health status of this group and identifying potential targets for intervention. The study design considers multiple psychological and physiological indicators, utilizes standardized assessment tools, and employs analytical procedures attentive to data distribution and covariate control, demonstrating methodological rigor. We offer the following suggestions for revision for the authors' consideration:
1.Please verify the official name of the affiliated institution listed as "University of Study of Bari".
2.Please remove the redundant notation "±" on Line 27 and Line 134 .
3.In the Abstract, the reported Spearman correlation coefficient (ρ) for the association between HGS and trait anxiety (ρ=−0.524) appears inconsistent with Table 3 (ρ=−0.532). Please correct the value in the Abstract to align with Table 3.
4.Please reconcile the FSS scoring description (Methods state mean score, cut-off ≥4) with the reported mean in Table 1 (36.19, implying sum score).
5.Please remove the redundant reporting of "(SD = ±8.96)" on Line 265.
6.Please perform a global check for the term "STAY-YI" (e.g., on Line 279). This appears to be a typographical error and should likely be "STAI-Y1" based on consistent usage elsewhere in the manuscript.
7.Please correct the spelling error "Famale" to "Female" in Table 2.
8.Please verify and correct the p-value reported on Line 300 for the correlation between physical activity and fatigue (p =0.011). This value appears inconsistent with the corresponding result in Table 3 (p<0.001).
9.Please correct the Spearman correlation coefficient (ρ) for HGS vs BDI reported on Line 304 and ensure consistency with Table 3, specifically regarding the decimal point (should be -0.524).
10.Table 4 requires reformatting for clarity, as negative signs and numerical values are sometimes split across lines or misaligned, making reading difficult. Specifically regarding the prediction of STAI-Y1 by "Self-reported weekly physical activity minutes," the reported unstandardized coefficient (β = -2.278) seems highly counterintuitive, although statistically non-significant (p=0.189). We recommend verifying this coefficient against the original data analysis output. Additionally, please ensure p-values in Table 4 are reported with greater precision where appropriate (e.g., p = 0.009 or p < .001) rather than solely as "0.000".
Author Response
Dear Reviewer,
Thank you very much for your thoughtful and constructive comments on our manuscript.
Please find below our detailed responses to each of your suggestions. We have addressed all points raised, and the corresponding revisions have been implemented in the manuscript.

Reviewer 2 Report
Comments and Suggestions for Authors
This study examined the relationships between self-reported physical inactivity, HGS, 24
and psychological distress, specifically depressive symptoms, anxiety, and cancer-related 25
fatigue (CRF), in physically inactive cancer survivors. While the sample size is small and research question being commonplace, the analyses seem thorough and robust.
Minor changes:
Line 464-469: Please discuss also exercise knowledge and attitude that should be measured along with practices. It is important to highlight that the amount of exercise (in practice) can be influenced by patient's knowledge and attitudes towards exercise; behavioural change requires just targeting practice but education and motivation. Suggest to cite this recent study by Prof Ray Chan - Hoe ZQ, Joseph R, Dick N, Thio CS, Wallen M, Chua LD, Miller C, Lee J, Chan RJ, Han CY. Nutrition and Exercise Knowledge, Attitude, and Practice: A Scoping Review of Assessment Questionnaires in Cancer Survivorship. Nutrients. 2025 Apr 23;17(9):1412.
Author Response
Dear reviewer,
Thank you very much for your careful reading of our manuscript and for your valuable suggestion regarding the Discussion section.
We fully agree with your comment and have incorporated your suggestion into the revised version of the manuscript.

Round 2
Reviewer 1 Report
Comments and Suggestions for Authors
The authors have satisfactorily revised the manuscript according to the suggestions made. The previous concerns have been well addressed, and the manuscript has been significantly improved. I now recommend its acceptance for publication.